# Scope of Artificial Intelligence in Screening and Diagnosis of Colorectal Cancer

**DOI:** 10.3390/jcm9103313

**Published:** 2020-10-15

**Authors:** Hemant Goyal, Rupinder Mann, Zainab Gandhi, Abhilash Perisetti, Aman Ali, Khizar Aman Ali, Neil Sharma, Shreyas Saligram, Benjamin Tharian, Sumant Inamdar

**Affiliations:** 1Department of Internal Medicine, The Wright Center for Graduate Medical Education, Scranton, PA 18505, USA; 2Saint Agnes Medical Center, Fresno, CA 93720, USA; rupindrmann@yahoo.com; 3Department of Medicine, Geisinger Community Medical Center, Scranton, PA 18510, USA; drzainabgandhi@gmail.com; 4Department of Gastroenterology and Hepatology, The University of Arkansas for Medical Sciences, Little Rock, AR 72205, USA; abhilash.perisetti@gmail.com; 5Division of Gastroenterology, The Commonwealth Medical College, Wilkes Barre General Hospital, Wilkes-Barre, PA 18764, USA; amanali786@hotmail.com; 6Digestive Care Associates, Kingston, PA 18704, USA; aminali92403@gmail.com; 7Division of Interventional Oncology & Surgical Endoscopy (IOSE), Parkview Cancer Institute, Fort Wayne, IN 46845, USA; neil.sharma@parkview.com; 8Division of Interventional Oncology & Surgical Endoscopy, Indiana University School of Medicine, Fort Wayne, IN 46805, USA; 9Department of Medicine, University of Texas Health San Antonio, San Antonio, TX 78229, USA; drsaligram@yahoo.com; 10General and Advanced Endoscopy, University of Arkansas for Medical Sciences, Little Rock, AR 72205, USA; Btharian@uams.edu; 11Advanced Endoscopy Fellowship, University of Arkansas for Medical Sciences, Little Rock, AR 72205, USA; Sinamdar@uams.edu

**Keywords:** artificial intelligence, colorectal cancer, colon cancer, polyp, screening, colonoscopy, computer-aided diagnosis

## Abstract

Globally, colorectal cancer is the third most diagnosed malignancy. It causes significant mortality and morbidity, which can be reduced by early diagnosis with an effective screening test. Integrating artificial intelligence (AI) and computer-aided detection (CAD) with screening methods has shown promising colorectal cancer screening results. AI could provide a “second look” for endoscopists to decrease the rate of missed polyps during a colonoscopy. It can also improve detection and characterization of polyps by integration with colonoscopy and various advanced endoscopic modalities such as magnifying narrow-band imaging, endocytoscopy, confocal endomicroscopy, laser-induced fluorescence spectroscopy, and magnifying chromoendoscopy. This descriptive review discusses various AI and CAD applications in colorectal cancer screening, polyp detection, and characterization.

## 1. Introduction

Artificial intelligence (AI) translates as having a computer program that mimics humans’ learning and problem-solving capability [1]. The concept of AI dates back to as early as Aristotle’s (384–322 BC) study of logic; however, Alan Turing (1912–1954) built the first operational computer in 1940 known as Electromechanical Heath Robinson [2]. AI in medicine has two main branches; virtual and physical. Machine Learning (ML) and Deep Learning (DL) represent the virtual branch of AI in medicine [3]. The ML involves computer statistics and analytics to make predictive and descriptive models by repeating specific tasks. The ML comprises of unsupervised and supervised learning. As the name implies, unsupervised ML is to feed the data without prior knowledge, and the machine identifies groups based on similarities in the data. In supervised ML, the machine is fed with input (individual descriptions) and output (an outcome of interest) data. The computer eventually outlines newer input/output pairs based on the information feeds [1]. Machine learning is broadly classified into supervised, unsupervised, reinforcement, and active learning tasks. Supervised learning involves input data with target labels to learn a pattern. Bayesian inferences, decision trees, linear discriminants, support vector machines, logistic regression, and artificial neural networks are different models of supervised ML [4]. Unsupervised learning involves recognizing patterns from the input data without previously defined target labels. Reinforcement learning involves training intelligent agents to enhance their performance [5].

A deep learning system accepts multiple data types as input which form layers of data, from which it extracts the data points of interest. Deep learning can be supervised or unsupervised. The most common models are trained using supervised learning, in which datasets are composed of input data and corresponding output data labels [6]. DL comprises of two steps: pre-training and fine-tuning. In step one, the DL model attempts to learn the underlying data distribution and creates outputs in an unsupervised manner. In step two, the output generated is tuned for the specific task at hand to achieve maximum performance [5]. DL is further classified into the deep neural network (DNN), recurrent neural network (RNN), and convolutional neural network (CNN). DNN is a feedforward method of artificial neural networks. CNNs are particularly important to identify patterns from image pixels with minimal pre-processing. RNNs are important for identifying sequential data in temporal sequence and thus work best for data such as time series [5]. The physical branch is the second branch of AI in medicine, including medical devices and robots [3]. Intensive research focused on using AI applications in the medical field is underway, which could provide unprecedented opportunities to improve the quality of healthcare [5].

Colorectal cancer (CRC) is the fourth most common cause of death due to cancer worldwide [7]. Screening colonoscopy resulted in a 70% decrease in CRC-related deaths by early detection and removal of preneoplastic and neoplastic lesions [8]. One of the most crucial colonoscopy parameters is the adenoma detection rate (ADR), a direct measure of the effectiveness of colonoscopy performed by an endoscopist [9]. Post-colonoscopy CRC (PC-CRC) and CRC-related mortality are inversely related to the adenomas detected during colonoscopy [9,10,11,12]. In addition, there are also various other tools, such as blood tests, stools teats, and imaging modalities, that aid in the screening for CRC [13].

The role of AI in gastroenterology is increasing rapidly, ranging from diagnosis and classification of dysplastic and neoplastic changes of the polyps to cystic fluid analysis and accurate prediction models to determine the need for intervention with computer-aided detection and diagnosis (CAD) [5,14]. The incorporation of AI and its tools with known methods of screening and diagnosis of CRC can increase the diagnostic accuracy and potentially attenuate CRC-related mortality. Additionally, by real-time differentiation of neoplastic and benign tumors, AI can also decrease the unnecessary removal of non-neoplastic tumors, reducing the overall cost, procedure time, and associated complications. In this descriptive review, we aim to discuss current advancements in the role of AI for CRC screening in various modalities, including colonoscopic examination, blood and stool testing, and imaging. We will also elaborate on the role of AI in the colorectal polyp detection and characterization.

## 2. Screening of Colorectal Cancer

CRC causes significant mortality and morbidity. Early diagnosis by effective screening methods has been shown to decrease both mortality and morbidity related to CRC. There are multiple screening methods for CRC screening. It includes invasive tests, such as colonoscopy and flexible sigmoidoscopy, minimally invasive tests, such as capsule endoscopy, stool tests, such as fecal immunochemical test (FIT), guaiac-based fecal occult blood test, and multitarget stool DNA (MTS-DNA), and radiologic tests, such as computed tomographic colonography [15,16].

### 2.1. Colonoscopy

Colonoscopy is considered the gold-standard CRC screening method due to high sensitivity, specificity, and ability to visualize directly and act (biopsy and resection of polyps) on cancerous and precancerous lesions. Multiple case-control studies and prospective studies have shown that colonoscopy screening resulted in a significant reduction in incidence and mortality due to CRC [17,18,19,20]. However, small-size or flat polyps could be missed by human eyes alone. A systematic review and meta-analysis showed a 22% pooled miss rate for polyps of any size with colonoscopy [11]. Another meta-analysis showed that 8.6% of CRC cases occur within three years of negative colonoscopy results [10]. With the recent advancement of AI, CAD systems, such as real-time automatic detection systems, have been studied to improve the ADR [21].

In an open-label, non-blinded, randomized study, 1058 patients were randomized to either routine colonoscopy (*n* = 536) or computer-aided detection (CADe), colonoscopy with real-time automatic polyp detection (*n* = 522), in order to determine the differences in polyp and adenoma detection rate in the two groups. The CADe group had a 1.89-fold higher mean number of polyps detected than the routine colonoscopy group (95% CI 1.63 vs. 2.192, *p* < 0.001). Similarly, the CADe group was also found to have higher a ADR (29.1% vs. 20.3%, *p* < 0.001) compared to the routine colonoscopy group. More importantly, this difference was more pronounced for diminutive adenomatous (185 vs. 102; *p* < 0.001) polyps. With CADe, the evidence suggests that an increased ADR is achieved compared with routine colonoscopy, leading to reduced colon cancer rates. For large adenomas, the mean ADR for CADe and routine colonoscopy group was (77 vs. 58, *p* = 0.075), not reaching statistical significance. However, this study was not double-blinded. Therefore, the results could have been confounded by potential unrecognized factors. Further, large multicenter double-blinded studies are needed [21].

### 2.2. Blood Tests

Certain demographic and blood test data can be used to identify patients with high risk for CRC. Complete blood count (CBC) is one of the laboratory tests that can help identify at-risk patients, especially due to the presence of microcytic iron deficiency anemia [22,23]. A retrospective case-control study revealed that colon cancer patients tend to have anemia and high red cell distribution width (RDW). High RDW was more pronounced in patients with right-sided colon cancer [24]. High RDW was also found to be associated with an increased risk of mortality in CRC patients [25]. In CRC patients, slow bleeding from cancer is thought to cause iron deficiency anemia and stools positive for the occult blood test, and it is observed more in elderly patients. AI can be useful in the risk assessment of the general population to determine individuals at high risk for developing CRC based on their demographic data, CBC, and age. Although this is a relatively newer method for estimating the risk of CRC, studies conducted have shown promising results [22,23].

A binational study used electronic medical record data from two unrelated patient populations (Israel and UK) to develop an ML-based prediction model (MeScore) for identifying individuals at high risk of CRC based on their CBC, age, and sex. The Israeli cohort data set was randomly divided into the derivation dataset (80%) and validation (20%) dataset, whereas the UK case-control data set was used to create an external validation set. The predictive model was created using the Israeli derivation dataset and then applied to both Israeli and UK validation datasets of CRC patients who had CBC done three to six months before the diagnosis. The area under the receiver operating characteristics curve (AUC) (measuring the overall performance of the model) for detecting CRC was 0.82 ± 0.01 and 0.81 for the Israeli and UK validation sets, respectively.

Similarly, specificity at 50% sensitivity was 88 ± 2% and 94 ± 1% for the Israeli and UK validation sets, respectively, for CRC cases 3–6 months before diagnosis. While comparing this model to age alone, AUC, Odds ratio, and specificity were 0.81 vs. 0.72, 32 vs. 2, and 90 vs. 79%, respectively, showing the predictive model’s outperformance. Similar results were obtained for the sex alone predictive value. Moreover, the number of detected new cases of CRC increased by 115% when gFOBT and predictive models were combined in the Israeli data set. This study showed that this model, MeScore, could detect high-risk patients in a primary care setting and potentially decrease the risk of developing colon cancer [23].

Kinar et al. conducted a study to analyze the performance of an ML-based algorithm (MeScore) in predicting CRC in average CRC risk individuals, based on their CBC [26]. A total of 112,584 subjects (aged between 50 to 75 years) with non-CRC from the Maccabi Health Service (MHS) dataset were recruited who underwent CBC in the last six months. Using the MeScore system, all these individuals were assigned a score from 1 to 100 based on the CBC report information. The model also incorporated demographic information such as the age and sex of the subjects as well. The average MeScore was found to be 59.3 and 46.8 in males and females, respectively. Using a MeScore cutoff of the top three percentile (score > 97.02) and top one percentile (score > 99.38), the odds ratio for CRC diagnosis was found to be 10.9 (95% CI 7.3 to 16.2) and 21.8 (95% CI 13.8 to 34.2), respectively. False-positive tests based on three percentile cutoffs were seen due to the use of anticoagulant, gastrointestinal illness, and another cancer diagnosis. This study showed that ML algorithm tools such as MeScore could be used to identify high-risk patients who should undergo screening colonoscopy [26].

The machine learning tool ColonFlag’s performance for early CRC detection was evaluated based on the gender, age, and CBC of a US-community-based insured population using the Kaiser Permanente Northwest Region tumor registry. A total of 17,095 patients were included in the study, with 900 CRC patients and 16,195 CRC-free control patients. Data about patient demographics and CBC was obtained from the dataset. The ColonFlag^®^ model was more sensitive in both age groups in detecting CRC cases among true CRC cases when compared to Hgb alone in the first and second 6 months after the CBC blood tests. It was also more sensitive in identifying CRC cases diagnosed in the first 180 days (39.9%) when compared to 181–360 days (27.4%) before CRC diagnosis, and detecting CRC cases between the 40–89-year-old CRC population age range when compared to the 50–75-year-old CRC population [22].

### 2.3. CT Colonography (CTC)

CTC is a non-invasive imaging test for CRC [27,28]. CAD can improve CTC diagnostic capability to differentiate between different lesions and improve detection capability. Song et al. conducted a study where they used the Haralick texture analysis method along with CTC to differentiate between various lesions based on texture features [28]. The virtual pathological model was formed based on the Haralick texture analysis method to investigate the usefulness of high order derivatives, such as gradient and curvature. Texture features were validated on 148 lesions of 8- to 30-mm sized polyps using a support vector machine classifier. The AUC of classification in differentiating neoplastic from non-neoplastic lesions improved from 0.74 (using the image intensity alone) to 0.85 (by combining the high-order texture features) [28].

A study was performed on the 24 patients who underwent colonoscopy and CTC on the same day and had non-polypoidal T1 tumors with the endoscopic classification of 0-IIa (*n* = 11) and IIa + IIc (*n* = 13). CAD software (ColonCAD API 4.0, Medicsight plc) was integrated with a CTC radiologic workstation. Data were collected at three sphericity settings, operating points for CAD, and analyzed using Fischer’s exact test. With CAD, tumor detection sensitivity increased as sphericity decreased (83.3, 70.8, and 54.1% at sphericity of 0, 0.75, and 1, respectively), whereas, false positive CADs per patient decreased with increasing sphericity. Thus, with CAD, there is increased accuracy with increasing sphericity. Small benign polyps led to false positive detection (over 20% at a sphericity setting of 0), although the majority were due to normal colon anatomy. The results of this study indicated that CAD could be useful for the detection of morphologically flat non-polypoidal cancer [27].

### 2.4. Colon Capsule Endoscopy

Colon capsule endoscopy (CCE) is a minimally invasive procedure that can be used as a CRC screening method in patients with incomplete colonoscopy and contraindication for sedation use. It requires more laxatives than colonoscopy and CTC as laxatives help expulsion of the capsule from the GI tract along with cleansing. Capsule endoscopy requires manual reading and interpretation of images for polyp detection, increasing the risk of error. AI techniques can automate the interpretation of results [29,30]. Balnes-Vidal et al. conducted a study to develop a DL-based convolutional neural network (CNN) algorithm for automatic polyp detection and also to develop an algorithm to match CCE- and colonoscopy-detected polyps based on their size, location, and morphology [29]. A total of 255 people who were FIT positive from the Danish national screening were included in this study. All these patients underwent first CCE and then colonoscopy and histopathology of removed polyps. Out of 255 patients, 131 had at least one polyp detected in both CCE and colonoscopy. A total of 168 polys were matched in both the CCE and colonoscopy groups by the polyp matching algorithm. The autonomous polyp detection algorithm showed accuracy, sensitivity, and specificity of 96.4, 97.1, and 93.33%, respectively, for polyp detection compared to manual polyp detection [29].

Artificial intelligence, in conjunction with various screening methods, has been shown to improve the screening of CRC and thus potentially decrease missed polyp rate. However, most of the studies use data that are not from prospective clinical trials so further randomized controlled studies are needed.

## 3. Polyp Detection

Although colonoscopy is considered a gold-standard test to detect CRC, it is not 100% sensitive. This is especially true for adenoma <5 mm. For adenoma that are 6 mm or larger, the sensitivity of colonoscopy ranges from 75 to 93% [31]. This sensitivity also depends on various factors, including bowel preparation, mucosal surface visibility, and operator dependency. Repeat surveillance colonoscopy is recommended in patients depending upon the number, histopathologic characteristics, and size of polyps [32]. A retrospective observation population-based analysis using National Health Service data from 2001 to 2010 showed that post-colonoscopy CRC (PC-CRC) rates ranged from 2.5 to 7.7%, depending upon the method used and exclusion criteria applied [10]. Another population-based study conducted on colonoscopy data and histopathological reports from the Netherlands cancer registry (2001–2010) showed that 86% of PC-CRCs were related to inadequate examination and missed or incomplete removed lesions [33]. Most of the missed PC-CRC were on the right-sided, proximal, flat, and small in size [10,33]. Therefore, there is a need for various methods/techniques to improve polyp detection to prevent PC-CRC and AI could be utilized to achieve this [7,32] (Table 1).

Karkanis et al. used the color wavelet features (CWF) technique to detect tumors from colonoscopic video frame sequences. Sixty patients with small polyps were included in this study, and results showed high sensitivity and specificity (99.3 ± 0.3% and 93.6 ± 0.8%, respectively) on classified image regions to detect colorectal polyps with the use of CWF features [38]. However, this study was conducted using static images instead of real-time colonoscopy videos [38]. Endoscopic imaging material classification can be done by either a pit-pattern scheme or coarse classification. There are six different classes based on pit-pattern and two different classes (benign and malignant) based on coarse classification. Hafner and colleagues described the use of an automated classification system for endoscopic images to detect tumors. A total of 484 zoomed-colonoscopic images were classified based on two/six different classes using discriminative frequency components. The classification accuracy for six and two classes was found to be 86.8% and 96.9%, respectively [39].

Urban et al. conducted a study which tested the ability of computer-assisted image analysis using CNN to detect polyps [40]. In this study, the CNN model was trained using a sample of 8641 images with 4088 unique polyps from more than 2000 screening colonoscopies. The training and testing of the CNN model were performed by different methods, such as cross-validation, training on the dataset, and testing on or against another expert reviewer as reference. CNN identified polyps with an accuracy of 96.4% and an AUC of 0.991 in cross-validation experiments. When tested on an independent dataset, AUC was found to be 0.74, with 96.4% accuracy. These results showed that the CNN model could decrease missed adenomas and, thus, improve ADR, but static images were used in this study. Therefore, further multicenter studies using live video are needed to evaluate the utility of CNN in colonoscopy [40]. There is high potential in the application of CNN to detect adenomas and screening of colorectal cancer.

Nevertheless, Fernandez-Esparrach et al. used routine colonoscopy videos to assess the capability of the Window Median Depth of Valley accumulation (WM-DOVA) energy maps system, which defines polyp boundaries as valleys of image intensity to overcome the challenge of static images [32]. Twenty-four videos containing 31 different polyps were taken from routine colonoscopies. With the WM-DOVA model, all polyps were detected correctly in at least one frame, but sensitivity was only 70%, using 3.75 as a threshold value for energy map maximum, which is likely due to a small study sample. This method was found to be more useful for detecting small and flat polyps, which are easy to miss [32]. In a pilot study, retrospective data were collected from a sample of 73 colonoscopy videos, including 155 colorectal polyps, to develop an AI-assisted CAD polyp detection system. Both frame-based and polyp-based analyses were performed. Sensitivity, specificity, and accuracy were found to be 90.0, 63.3, and 76.5%, respectively, for frame-based analysis, whereas for polyp-based analysis, sensitivity was found to be 94% [36].

To further study the efficacy of polyp detection in real-time, a DL algorithm was developed with data from 5545 colonoscopy images of 1290 patients [35]. Image analysis validation was done on 27,113 colonoscopy images obtained from 1138 patients with at least one detected polyp (Dataset A) and a public database of 612 images from 128 colonoscopy videos with confirmed polyps (Dataset B), whereas video analysis validation was done on videos of 38 colonoscopies with 110 confirmed polyps (Dataset C) and also on full-length unalerted colonoscopy videos from 54 patients (Dataset D). For Dataset A, per-image sensitivity, specificity, and AUC was 94.38% (95% confidence interval (CI): 93.80, 94.96%), 95.92% (95% CI: 95.66, 96.18%), and 0.984, respectively, and for Dataset B, per-image sensitivity was 88.24% (95% CI: 85.76, 90.72%). For Dataset C, per-image sensitivity was found to be 91.64% (95% CI: 91.42, 91.86%) and per-polyp sensitivity was 100%, and for Dataset C, per-image specificity was 95.40% (95% CI: 95.36, 95.44%). The use of a multi-threaded processing system in the algorithm can process 25 frames per second, and the latency was of 76.80 ± 5.60 milliseconds in real-time video analysis. This CAD system is shown to have high performance both in image and video colonoscopy and can be used as a quality measure and also aid endoscopists in diagnosis by providing concurrent objective aspects during colonoscopy [35].

CAD-based systems, especially deep learning techniques, are promising options to decrease human variation by providing real-time support and, thus, assisting in polyp detection. Most of these systems are studied in small studies, often with static imaging. Further randomized studies are needed to validate these results and evaluate the effects of other quality and operator parameters, such as bowel preparation, colonoscope withdrawal time, and quality of inspection, which can affect CAD polyp detection performance.

## 4. Polyp Characterization

Another critical aspect in the diagnosis of colorectal polyp is accurate polyp characterization. Most of the current literature differentiates polyps into neoplastic or non-neoplastic polyps. Artificial intelligence for polyp characterization may have several potential advantages, such as predicting malignant vs. benign lesions, improving the endoscopist learning phase, and guiding endoscopic therapy for submucosal tumors [41] (Table 2). While most small colorectal polyps are hyperplastic with little to no risk of turning to CRC, an accurate diagnosis of these polyps is needed to prevent unnecessary resections and complications associated with it. In a single-center study from Japan, a unique CNN system based on CAD utilizing AI was developed to study endoscopic images extracted from colonoscopy videos [42]. A total of 1200 images from colonoscopies were included, and additional video images from 10 cases were applied as a test. The accuracy of the 10-fold cross-validation test was found to be 0.751, meaning the decision by CNN was correct in 7 out of 10 cases [42].

Various methods of polyp characterization have been developed. Conventional white-light endoscopy is a widely available endoscopic modality that uses the full visible wavelength range (400–700 nm) [43]. A retrospective study was conducted at three hospitals to develop and assess deep learning models for the automatic classification of colorectal lesions histologically in white-light colonoscopy images. A total of 3828 images from 1339 patients were included in this study. Images were divided into seven categories based on pathologic results and then reclassified into four categories. Two CNN architectures, ResNet-152 and Inception-ResNet-v2, were used to consult deep learning models. The mean accuracy for seven-class classification by ResNet-152 and Inception-ResNet-v2 was 60.2 and 56.4%, respectively, in the internal test dataset and 74.7 and 74.3%, respectively, in external datasets. Similarly, the mean accuracy for the four-class classification by ResNet-152 and Inception-ResNet-v2 was 67.3 and 67.7%, respectively, in the internal test dataset and 79.2 and 76.0%, respectively, in external datasets. The mean AUC of the better performing model, Inception-ResNet-v2, was 0.832 and 0.935 for tubular adenoma with or without low-grade dysplasia and high-grade dysplasia colorectal lesions, respectively [44]. This study’s results showed promising results of deep learning models in the classification of various colorectal lesions. It can easily be adopted into clinical practice given that white-light endoscopy is widely used, and it requires no additional training compared to other advanced image-enhanced endoscopy techniques.

Multiple advanced modalities, such as magnifying narrow-band imaging (NBI), endocytoscopy, confocal endomicroscopy, laser-induced fluorescence spectroscopy, and magnifying chromoendoscopy, are being utilized in polyp characterization.

### 4.1. Magnifying Narrow Band Imaging (NBI)

NBI is an image-enhanced form of endoscopy that narrows the spectral transmittance bandwidth using optic filters for sequential green and blue illumination. This technique helps examine microvascular patterns associated with histological features and depth of submucosal invasion. NBI magnification can be useful for polyp characterization [47,62].

In a prospective study, the computer-based method was developed for the classification of colorectal polyps [44]. A total of 214 patients with 434 polyps of size 10 mm or less were included in the study, and all these patients underwent zoomed NBI colonoscopy. The diagnostic performance of two experts (who routinely used magnification colonoscopy with NBI for >4 yrs), two non-experts (performing colonoscopy for at least one year but never used NBI), and a computer-based algorithm was compared for polyp classification as neoplastic or non-neoplastic. Results for polyp classification were comparable for expert and computer-based methods with sensitivity, specificity, and accuracy of 93.4 vs. 95.0%, 91.8 vs. 90.3%, and 92.7 vs. 93.1%, respectively, whereas the non-expert group was inferior with a sensitivity of 86.0%, a specificity of 87.8%, and an accuracy of 86.8% [46]. In another retrospective study, a computer-aided system was developed to predict the classification of colorectal lesions based on NBI magnifying colonoscopy images. A total of 371 NBI magnifying images of colorectal lesions from patients who underwent colonoscopy between January 2005 to July 2010 were included, and the performance of the computer-aided system was compared to two experienced endoscopists and histologic diagnosis. The computer-aided system showed diagnostic accuracy, sensitivity, and specificity of 97.8, 97.8, and 97.9%, respectively. Interestingly, the diagnostic agreement between computer-aided classifier systems and two experts was 98.7%, with no significant difference [47].

The optical diagnosis of colorectal polyps differs between endoscopists, so a computer-assisted optical biopsy (COAB) system was developed using machine learning to differentiate between neoplastic and non-neoplastic polyps. A total of 275 polyps were detected during colonoscopy and imaged using the unmagnified high-definition white light and narrowband image mode [62]. Two experts also reviewed a total of 788 images available (602 were for training machine learning algorithms and 186 for COAB testing) and all images in optical polyp characterization. The CAOB approach’s accuracy, sensitivity, and negative predictive value were 78.0, 92.3, and 88.2%, respectively. However, the accuracy obtained by two expert endoscopists was 84.0% (*p* = 0.307) and 77.0% (*p* = 1.000) and thus did not differ significantly from COAB predictions [63]. The studies mentioned above indicate that computer-assisted NBI image analysis may play a role in polyp characterization during colonoscopy.

Byrne et al. developed a deep convolutional neural network (DCCN) for real-time assessment of untouched endoscopic video images to differentiate between adenomatous and hyperplastic diminutive colorectal polyps [47]. Only NBI video frames were used in this study. Out of 125 polyp videos evaluated by the AI model to differentiate between adenomatous and hyperplastic polyp, it did not build confidence to predict histology in 19 polyp videos. In the remaining 106 videos with high confidence for prediction, AI model accuracy, sensitivity and specificity for identification of adenoma were 94 (95% CI: 86 to 97%), 98 (95% CI: 92 to 100%), and 83% (95% CI: 67 to 93%), respectively. This model showed high accuracy in differentiating adenomatous and hyperplastic polyps. Although this model showed promising results, these results need to be validated in true live colonoscopies [61]. Another study conducted where a CAD with deep neural network (DNN-CAD) was developed and tested to classify the diminutive colorectal polyp NBI images [46]. From Taiwan’s tertiary hospital database, 1476 images of neoplastic polyp images and 681 hyperplastic polyp images with a size less than 5 mm were obtained. Histology information of all these polyps was used as a reference. Information from a test set of 96 hyperplastic and 199 neoplastic polyps’ images was used to compare the diagnostic ability of DNN-CAD with novice (*n* = 4) and expert endoscopists (*n* = 2). DNN-CAD showed sensitivity, specificity, positive predictive value (PPV), and negative predictive value (NPV) of 96.3, 78.1, 89.6, and 91.5%, respectively, in identifying hyperplastic and neoplastic polyps in a test set. The average time to classify polyps by DNN-CAD was 0.45 ± 0.07 s, while that for experts was 1.54 ± 1.30 s and for non-experts was 1.77 ± 1.37 s (*p* < 0.001). The intraobserver agreement (kappa score) was 1 in DNN-CAD-classified polyps; however, the intraobserver and interobserver agreement among the novice and expert endoscopists were lower [55].

The use of narrow-band imaging along with computer assistance can help determine the characteristic of the polyps, which is equivalent to an expert in these preliminary studies. Further studies are needed to explore the enhancement of the diagnostic ability to improve polyp characterization.

### 4.2. Magnifying Chromoendoscopy

Magnifying chromoendoscopy is a technique to enhance visualization of pit patterns of polyp surfaces to differentiate between benign and neoplastic polyps. Magnification endoscopes have an individual lens attached, increasing the magnifying factor from 80 to up to 150 and improving accuracy in detecting lesions during colonoscopy. In magnifying chromoendoscopy, dye spray, such as indigo carmine or methylene blue/crystal violet, is used along with magnifying endoscopy to recognize the pit patterns on the polyp surface [64,65,66]. Pit pattern can be very helpful in the diagnosis of submucosal CRC [64]. CAD can be used with magnifying chromoendoscopy to automate and diagnose the malignant potential of the colorectal polyps with high sensitivity [65,66]. There can be interobserver variation even between experienced endoscopists to completely characterize the pit pattern correctly.

Takemura and colleagues created and analyzed an automated computer-based system, named HuPAS version 1.3, that could outline pits identified on digital magnifying endoscopic images [61]. A total of 134 regular pit pattern images were included in the study to compare the ability of an automated computer-based system and an endoscopist to characterize colorectal polyps. The automated computer-based system showed an accuracy of 98.5% (132/134) in identifying the colorectal lesion’s pit patterns. Their computer-based system was in 100% agreement with endoscopic diagnosis by the endoscopist for type I and II pit patterns, and for type IIIL and IV, it was able to diagnose in 96.6 and 96.7% cases, respectively [65].

### 4.3. Endocytoscopy

Endocytoscopy (EC) allows ultra-magnification of the real-time images by 380- to 500-fold. In EC, a contact light microscopy system is added to the colonoscope’s distal tip, enabling on-site evaluation of nuclei and cytological structures for pathologic diagnosis of lesions in real-time. It is shown to be 94.1% accurate in differentiating neoplastic lesions compared to 96% with biopsy [50,64]. EC requires expert endoscopists to interpret results, and it is very operator-dependent. Therefore, developing a CAD system for EC will allow interpretation by non-expert endoscopists also.

CAD-EC was developed and tested in a pilot study of 152 patients with small colorectal polyps (≤10 mm) [64]. The CAD-EC system was compared with two experts and two trainee endoscopists in predicting neoplastic changes in the colorectal lesions. For CAD-EC and EC evaluation by experts, the accuracy and sensitivity were found to be 89.2 (95% CI, 83.7–93.4%) vs. 92.3% (95% CI, 89.0–94.9%) and 92 (95% CI, 86.1–95.9%) vs. 92.7% (95% CI, 89.0–95.5%), respectively. Trainee endoscopists had a much lower accuracy of 80.4% and a sensitivity of 81.8% when compared to CAD-EC. The results of this study showed that CAD-EC has a sensitivity and accuracy comparable to expert endoscopists and could also provide an instant diagnosis as it takes 0.3 s per lesion [50].

In another retrospective study, CAD-EC was developed and evaluated to differentiate between invasive CRC and adenomatous lesions [65]. The image database was generated based on a consecutive series of EC images from 242 patients. From the dataset, 5543 images were used to construct the CAD-EC algorithm and 200 images to test the system. CAD-EC showed a sensitivity of 89.4%, a specificity of 98.9%, an accuracy of 94.1%, a PPV of 98.8%, and an NPV of 90.1% in differentiating invasive cancer from adenoma. Although this study showed a promising result, it was conducted on a database of EC images, therefore further multicenter clinical trials on real-time colonoscopy videos are needed [67].

### 4.4. Confocal Endomicroscopy/Confocal Laser Endomicroscopy

Confocal endomicroscopy/confocal laser endomicroscopy has been available in the biomedical field since 1961 [66]. It allows magnification up to 1000-fold in real-time images and allows real-time in vivo histological images of gastrointestinal mucosa [68]. Probe-based confocal laser endomicroscopy (pCLE) is usually performed by endoscopist experts in this technique and requires much training. A computer-based system can provide objective support for pCLE diagnosis. André et al. developed and studied the diagnosis ability of computer-based automated pCLE classification and compared it with expert endoscopists who made a diagnosis based on pCLE videos alone to differentiate between neoplastic and non-neoplastic polyps [50]. A total of 135 images of colon polyps from 76 patients were included, and histopathological diagnosis was used as a standard criterion. The results revealed a sensitivity, specificity, and accuracy of 92.5 vs. 91.4%, 83.3 vs. 84.7%, and 89.6 vs. 89.6%, respectively, for computer-based automated pCLE classification and expert performance in differentiating neoplastic and non-neoplastic lesions, and these differences were not statistically significant [48]. Thus, the development of confocal laser endomicroscopy with computer assistance shows promise for polyp characterization; however, widespread use needs further evaluation with randomized controlled studies.

### 4.5. Laser-Induced Fluorescence Spectroscopy (LIFS)

Laser-induced fluorescence spectroscopy (LIFS) is a technique that provides real-time automatic differentiation of colorectal polyps as benign or neoplastic. It includes an optical fiber device, WavSTAT, that is installed into biopsy forceps. It emits laser waves that are absorbed by targeted tissue and then releases light to give optical biopsy results of whether the targeted lesion is neoplastic or non-neoplastic with the help of a computer software algorithm [49,69]. Diagnostic accuracy of WavSTAT was compared to WavSTAT along with high-resolution endoscopy in 87 patients with 207 colorectal polyps (size less than 10 mm). The diagnostic accuracy of WavSTAT alone (74.4%) and WavSTAT with high-resolution endoscopy (79.2%) did not meet the criteria for the American Society for Gastrointestinal Endoscopy (ASGE) performance threshold for assessment of small colorectal lesions [69]. Rath et al. studied a new version of the LIFS system called WavSTAT4, which could predict a colorectal neoplasm in vivo within 1 sec. In a prospective observational study, histology of 137 small polyps (≤5 mm size) from 27 patients who underwent screening or surveillance colonoscopy was predicted with LIFS using WavSTAT4 and compared to traditional histopathological results [49]. The accuracy of LIFS using WavSTAT4 in predicting polyp histology was 84.7%. Meanwhile, for distal colorectal diminutive polyps only, the NPV for excluding adenomatous histology increased to 100%. This study shows that LIFS with WavSTAT4 can predict colorectal lesion histology, and these results were more pronounced in small polyps in the distal colorectal area [49].

### 4.6. Autofluorescence Endoscopy

Endogenous fluorophores in the colorectal tissue emit natural tissue fluorescence upon excitation by light. The autofluorescence imaging (AFI) system analyzes fluorescence and provides a green/red (G/R) image [8]. AFI colonoscopy produces real-time pseudo-color images with neoplastic lesion appearing green and non-neoplastic lesion appearing red/magenta [70,71]. Aihara and colleagues developed a color analysis software that enables analysis of colorectal lesion with AFI and studied the diagnostic ability of this software in a prospective study of 32 patients with 102 colorectal lesions. Lesions were labeled as neoplastic (<1.01) and non-neoplastic (≥1.01) based on the G/R ratio. Results showed that the mean G/R ratio was 0.86 for neoplastic lesions, 1.12 for non-neoplastic lesions, and 1.36 for normal mucosa. This study showed that colorectal lesions could be differentiated into neoplastic and non-neoplastic based on AFI and decrease unnecessary interventions [70].

Although there are CAD-based systems for polyp characterization produced from white-light endoscopy, most of them are developed to use with advanced imaging modalities. AI helps predict malignant and non-malignant tissue to guide therapy, but most of the evidence is based on small studies. No randomized trial has been done, so large randomized trials are needed to validate these results.

## 5. Conclusions

AI has expanded exponentially over the last few years in the gastroenterology field, especially in gastrointestinal cancer screening and diagnosis. Implementation of AI and CAD technology with colonoscopy and various endoscopic modalities is showing promising results for screening and diagnosis of CRC. In colorectal polyp detection and classification, AI and CAD can provide clinicians with assistance in establishing diagnosis by providing concurrent objective aspects. Multiple studies showed that computer-aided software could provide real-time optic biopsies comparable to expert endoscopist performances; however, several study limitations need to be kept in mind, such as most of the data being available based on small studies at tertiary care centers, selection bias of the images used for the training set, lack of randomization of many studies, and the number of images used for the AI model’s training set. Preliminary results with small studies have shown promising results, and further large, multicenter clinical trials are needed to establish the diagnostic accuracy of AI technology in the real world.

## Figures and Tables

**Table 1 jcm-09-03313-t001:** Studies on artificial intelligence (AI) in polyp detection.

Author, Year, and Reference	Dataset	AI System	Imaging Modality	Results	Conclusion	Limitations
Fernandez-Esparrach et al., 2016 [32]	24 colonoscopy videos containing 31 different polyps	Window Median Depth of Valleys Accumulation (WM-DOVA) energy maps	WLI	All polyps from 24 colonoscopy videos were detected in at least one frame. The mean of the maximum values on the energy map was higher for frames with polyps than without (*p* < 0.001). Performance improved in high quality frames (AUC = 0.79 (95% CI 0.70–0.87) vs. 0.75 (95% CI 0.66–0.83)).	It showed WM-DOVA maps as one of potential method for an accurate polyp detection tool.	In some cases, lateral observation of polyps led to detection errors due to presence of other elements in scene with valley formation (blood vessels and fold)
Park and Sargent, 2016 [34]	11,802 image patches extracted from 35 colonoscopy videos	CNN	WLI and NBI conditional random field (CRF) model	Images were classified using a CRF model for colonoscopic polyp detection and showed method had 86% sensitivity and 85% specificity when evaluated on a feature training set of 11,802 images from 35 colonoscopy videos with accompanying endoscopy reports.	The CNN-derived features showed great invariance to viewing angles and image quality factors when comparted to the eigenimage model.	Feature relationships in adjacent video frames were not fully incorporated into CNN.
Misawa et al., 2018 [35]	73 colonoscopy withdrawal videos containing 155 polyps (1.8 million total frames)	CNN	WLI	The sensitivity, specificity, and accuracy for the frame-based analysis, were 90.0, 63.3, and 76.5%, respectively.	This study showed that AI has the potential to provide automated detection of colorectal polyps.	It is a retrospective study so further prospective studies needed.
Urban et al., 2018 [36]	Image dataset: 8641 images from >2000 patients (4088 polyp and 4553 non-polyp); 2 video sets: 20 videos (10 in each set), 28 and 73 polyps in 1st and 2nd video set, respectively	CNN	WLI	Image dataset: the CNN identified polyps with an AUC of 0.991 and an accuracy of 96.4%. Video dataset: expert reviewers identified 8 additional polyps that had not been removed without CNN assistance and an additional 17 polyps with CNN assistance.	Showed that this system could increase ADR and reduce interval colorectal cancers.	Requires validation of these results in large multicenter trials as it is based on single-center study.
Figueiredo et al., 2019 [37]	42 patients; 1680 polyps instances. 1360 normal mucosa frames	SVM binary classifiers	WLI	There are three methods used in this study and all are binary classifiers, labeling a frame as either containing a polyp or not. Two methods (methods 1 and 2) are threshold-based, and method 3 belongs to the machine learning class. The sensitivity, specificity and accuracy were found to be 83.7 vs. 61.6 vs. 99.7%, 66.6 vs. 61.3 vs. 79.6%, and 74.3 vs. 63.2 vs. 90.1 for method 1, 2, and 3, respectively.	CAD showed good accuracy in the detection of polyps with white-light colonoscopy using all three methods.	Algorithm was not studied in real-time.
Wang et al., 2019 [21]	1058 patients; 53 colonoscopy videos (22 polyp, 31 non-polyp videos)	CNN	WLI	Significantly increased ADR (29.1 vs, 20.3%, *p* < 0.001) and the mean number of adenomas per patient (0.53 vs. 0.31, *p* < 0.001).	In a low prevalent ADR population, an automated polyp detection system leads to significant increases in both colorectal polyp and adenoma detection rates.	There was no external validation of study results. Unexpectedly, there were false positives in the system which were likely due to detection of medication capsules, of local sites of bleeding, or of undigested debris causing distractions during procedure.

CNN—convolutional neural network; WLI—white-light imaging; NBI—narrow band imaging; SVM—support vector machine; ADR—adenoma detection rate; AUC—area under the receiver operating characteristics curve; NPV—negative predictive value; PPV—positive predictive value.

**Table 2 jcm-09-03313-t002:** Studies on AI in polyp characterization.

Author, Year and Reference	Dataset	AI System	Imaging Modality	Results	Conclusion	Limitations
Tischendorf et al., 2010 [45]	209 polyps (160 neoplastic and 49 non-neoplastic) from 128 patients	Region growing algorithm	Magnification NBI	The sensitivity and specificity for polyp detection was 93.8 vs. 90% and 85.7 vs. 70% for human observer and computer-based approach, respectively.	Although automatic colon polyp classification is possible using NBI vascularization features, it is inferior to human experts.	
Gross et al., 2011 [46]	434 polyps (258 neoplastic and 176 non-neoplastic) from 214 patients	Computer-based algorithm	Magnification NBI	Sensitivity, specificity, and accuracy for polyp detection was found to be 93.4 vs. 95.0 vs. 86.0%, 91.8 vs. 90.3 vs. 87.8%, 92.7 vs. 93.1 vs. 86.8% for the expert group, computer-based algorithm, and non-expert group, respectively.	Computer-based algorithm results were found to be comparable to expert group and superior to non-expert group.	Although the computer-based algorithm showed high diagnostic, it is still not a fully automatic classification system.
Takemura et al., 2012 [47]	134 pit pattern images	SVM	Magnification chromoendoscopy	Diagnostic concordance between the computer-aided classification system and the two experienced endoscopists was 98.7% (366/371).	This study showed that computer-aided system is reliable for predicting the histology of colorectal tumors and there is no significant difference in diagnosis ability of a computer-aided system and an experienced endoscopist.	It is a retrospective, single-center study.
André et al., 2012 [48]	135 colorectal lesions (93 neoplastic and 42 non-neoplastic) in 71 patients	Retrieval-based software classification	Confocal laser endomicroscopy	The accuracy, sensitivity, and specificity were 89.6 vs. 89.6%, 92.5 vs. 91.4%, and 83.3 vs. 85.7% for automated probe-based confocal laser endomicroscopy (pCLE). software classification. and two expert endoscopists, respectively, with no statistically significant difference in the performance.	The automated pCLE software classification achieved higher performance than the off-line diagnosis of pCLE videos established by expert endoscopists.	A large training database is needed to be adequately representative of non-typical pCLE cases. The biopsy may be taken accidentally from the area that does not correspond with the obtained pCLE imaging.
Rath et al., 2015 [49]	137 diminutive colorectal polyps in 27 patients	WavSTAT4	Laser-induced fluorescence spectroscopy	For predicting polyp histology, LIFS using WavSTAT4 had an overall accuracy of 84.7%, sensitivity of 81.8%, specificity of 85.2%, and NPV of 96.1%. For distal colorectal diminutive polyps only after excluding adenomatous histology, the overall accuracy was 82.4%, sensitivity was 100%, specificity was 80.6%, and increase in NPV to 100%.	This study showed that LIFS using the WavSTAT4 system works precisely enough to support leaving distal colorectal polyps in place.	It is a single-center study. Patients in this study had more than one polyp and it cannot be excluded that these clustered observations might have biased the results of the study to some extent.
Mori et al., 2015 [50]	176 polyps (137 neoplastic and 39 non-neoplastic) from 152 patients	Support vector machine	Endocytoscopy	EC-CAD had a sensitivity of 92.0% and an accuracy of 89.2%; these were comparable to those achieved by expert endoscopists (92.7% and 92.3%; *p* = 0.868 and 0.256, respectively) and significantly higher than those achieved by trainee endoscopists (81.8% and 80.4%; *p* < 0.001 and 0.002, respectively)	EC-CAD provides fully automated instant classification of colorectal polyps with excellent sensitivity, accuracy, and objectivity.	This study used still images instead of real-time analysis for EC-CAD, which may have skewed results in favor of EC-CAD.
Mori et al., 2016 [51]	205 polyps (147 neoplastic and 58 non-neoplastic) from 123 patients	Support vector machine	Endocytoscopy	CAD was accurate for 89% of diminutive polyps and 89% of small polyps, which was comparable with the experts’ results (90%, *p* = 0.703; and 91%, *p* = 0.106, respectively) and significantly higher than results for the non-experts (73%, *p* < 0.001; and 76%, *p* < 0.001, respectively)	CAD application in endocytoscopy can be helpful in the management of diminutive/small colorectal polyps.	The web-based test diagnoses were based on only high-quality images. This can lead to bias as most of the routine endocytoscopies are not performed by experts.
Kominami et al., 2016 [52]	118 colorectal lesions (73 neoplastic and 45 non-neoplastic) from 41 patients	SVM	Magnification NBI	Concordance between the endoscopic diagnosis and diagnosis by a real-time image recognition system with a SVM output value was 97.5% (115/118). Accuracy between the histologic findings of diminutive colorectal lesions (polyps) and diagnosis by a real-time image recognition system with a support vector machine output value was 93.2%	This real-time image recognition system may fulfill The Preservation and Incorporation of Valuable Endoscopic Innovations (PIVI) recommendations and helpful in predicting the histology of colorectal tumors.	It requires magnifying colonoscopy, which needs extra training.
Misawa et al., 2016 [53]	100 images (50 neoplastic and 50 non-neoplastic) 173 images	CAD	Endocytoscopy with NBI (EC-NBI)	In this study, the CAD system provided a diagnosis for 100% (100/100) of the validation samples with a diagnosis time of 0.3 s per image. The diagnostic accuracy for adenomatous lesions is 90% with sensitivity, specificity, accuracy, PPV, and NPV of 84.5, 97.6, 98.0 and 82.0%, respectively.	This CAD system provides a fully automated computer diagnosis without the need for any dye solution.	It cannot diagnose cancers and sessile serrated adenomas/polyps (SSA/Ps) because there are currently few EC-NBI images of invasive cancers and SSA/Ps for training.
Mesejo et al., 2016 [54]	76 videos (40 adenomas, 21 hyperplastic lesions, and 15 serrated adenomas)	Combined machine learning and computer vision algorithms	WLI and NBI		This system usually performed better than human operators (including experts). It correctly classified more serrated adenomas and adenomas while keeping similar accuracy in terms of hyperplastic lesions.	
Chen et al., 2018 [55]	284 diminutives polyps (188 neoplastic and 96 hyperplastic) from 193 patients	Deep learning algorithm, CNN	Magnification NBI	In the test set, the DNN-CAD identified neoplastic or hyperplastic polyps with 96.3% sensitivity, 78.1% specificity, a PPV of 89.6%, and an NPV of 91.5%. DNN-CAD classified polyps as neoplastic or hyperplastic in 0.45 ± 0.07 s-shorter than the time required by experts (1.54 ± 1.30 s) and non-experts (1.77 ± 1.37 s) (both *p* < 0.001).	DNN-CAD provides accurate and consistent diagnostic performance for colorectal polyps and is not inferior to experts in the field.	The DNN-CAD diagnosis was based on high-quality images, and bias might occur with poor-quality images.
Mori et al., 2018 [56]	466 polyps from 325 patients in 18 centers of Japan	CAD, Support vector machine	NBI	Overall, 466 diminutive (including 250 rectosigmoid) polyps from 325 patients were assessed by CAD, with a pathologic prediction rate of 98.1% (457 of 466).	The real-time use of the fully automated CAD system designed for endocytoscopies can meet the clinical threshold required for the diagnose-and leave strategy for diminutive, non-neoplastic rectosigmoid polyps. This can help to improve the cost-effectiveness of colonoscopy.	It is a single-center study and no comparative data available.
Min et al., 2019 [57]	217 polyps from 91 patients were included as the test set. Of these polyps, 36 were excluded due to lost histopathology	Gaussian mixture model	Linked-color imaging	The accuracy of the CAD system was comparable to that of the expert endoscopists (78.4% vs. 79.6%; *p* = 0.517).The diagnostic accuracy of the novices endoscopist was significantly lower to the performance of the experts (70.7% vs. 79.6%; *p* = 0.018).	This novel CAD system developed based on linked-color imaging demonstrates a promising performance and is comparable to the expert endoscopist.	This study was performed using still images rather than real-time evaluations of polyps.
Sánchez-Montes et al., 2019 [58]	Images of 225 polyps were evaluated (142 dysplastic and 83 nondysplastic)	Support vector machines	WLI	The CAD system correctly classified 205 polyps (91.1%): 131/142 dysplastic (92.3%) and 74/83 (89.2%) nondysplastic. For the subgroup of 100 diminutive polyps (≤5 mm), CAD correctly classified 87 polyps (87.0%): 43/50 (86.0%) dysplastic and 44/50 (88.0%) nondysplastic. There were no statistically significant differences in polyp histology prediction between the CAD system and endoscopist assessment.	This computer vision system, based on the characterization of the polyp surface in white light, accurately predicted colorectal polyp histology.	Sessile serrated polyps were not included as a separate group because they are not considered as a different group in the Kudo and NICE classifications.
Horiuchi et al., 2019 [59]	Ninety-five patients with 429 polyps (258 diminutive rectosigmoid polyps and 171 diminutive non-rectosigmoid polyps)	Color intensity analysis software	Autofluorescence imaging	The accuracy, sensitivity, specificity, and PPV for differentiating diminutive rectosigmoid neoplastic polyps by CAD-AFI were 91.5, 80.0, 95.3, and 85.2%, respectively. For diminutive rectosigmoid polyps, the NPV for differentiating neoplastic polyps was 93.4% (184/197) with CAD-AFI and 94.9% (185/195) with trimodal imaging endoscopy.	This study suggests that CAD-AFI is an effective and objective modality for differentiating adenomatous from non-neoplastic rectosigmoid polyps.	It is a single-center study. It included patients who had known colon polyps.
Kudo et al., 2019 [60]	69,142 endocytoscopic images, taken at 520-fold magnification from 2000 polyps	EndoBRAIN, an artificial intelligence-based system	Endocytoscopy with narrow-band imaging	In the analysis of stained endocytoscopic images, EndoBRAIN identified colon lesions with a sensitivity of 96.9%, specificity of 100%, an accuracy of 98%, a PPV of 100%, and an NPV of 94.6%, and these values were all significantly greater than those of the endoscopy trainees and experts. In the analysis of narrow-band images, EndoBRAIN distinguished neoplastic from non-neoplastic lesions with a sensitivity of 96.9%, a specificity of 94.3%, an accuracy of 96.0%, a PPV of 96.9%, and an NPV of 94.3% and these values were all significantly higher than those of the endoscopy trainees; sensitivity and NPV were significantly higher, but the other values are comparable with experts.	EndoBRAIN accurately differentiated neoplastic from non-neoplastic lesions in stained endocytoscopic images and endocytoscopic narrow-band images, with histopathology used as the standard.	The web-based test diagnoses were made using 326 only high-quality images, which can cause bias as most of the endocytoscopies are not performed by experts.
Byrne et al., 2019 [61]	125 diminutive polyp videos (74 adenomas and 51 hyperplastic polyps)	Deep convolutional neural network	NBI	The AI model did not generate sufficient confidence to predict the histology of 19 diminutive polyps in the test set. For the remaining 106 diminutive polyps, the accuracy, sensitivity, specificity, NPV, and PPV for identification of adenomas of the model were 94, 98, 83, 97, and 90%, respectively.	An AI model trained on endoscopic video can differentiate diminutive adenomas from hyperplastic polyps with high accuracy.	The study used video recordings rather than real-time assessments of polyps

CNN—convolutional neural network; WLI—white-light imaging; NBI—narrow band imaging; SVM—support vector machine; CAD—computer-aided detection; NPV—negative predictive value; PPV—positive predictive value; CAD-AFI—computer-aided diagnosis using autofluorescence imaging; EC-CAD—computer-aided diagnostic system for endocytoscopic imaging.

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
