# Peer review of "Scope of Artificial Intelligence in Screening and Diagnosis of Colorectal Cancer"

_jcm, 2020, doi:10.3390/jcm9103313_

Round 1
Reviewer 1 Report
To the authors:
- Deep learning is not clearly described as a separate branch of machine learning that involve neural networks - these can be used for either supervised or unsupervised learning (e.g. autoencoders).
- Minor grammatical changes throughout that would benefit from a skilled editor: examples (not limited to these) include awkward phrasing with Line 114 "However, the small size or flat polyps could be missed by the naked eyes." - should be "could be missed by the human eye alone" or "could be missed by the naked eye". Line 126: "Thus with CADe, the risk of developing colon cancer can decrease as the ADR increases when compared with routine colonoscopy." should be more clear with "With CADe, the evidence suggests that increased ADR is achieved compared with routine colonoscopy, which may lead to reduced rates of colon cancer." ADR is a proxy measurement, and does NOT mean that the rates of colon cancer will decrease. Line 219 " However, most of the data available from small studies, so further randomized controlled studies are needed." Should be corrected with a verb, "most of the studies use data that are not from prospective clinical trials..." Line 288 is a run-on sentence. Line 295 sentence does not have a verb.
- The significance of results in context are still not clearly changed. For example, Line 155 "While comparing this model to age alone, AUC, Odds ratio, and specificity were 0.81 vs. 0.72, 32 vs. 2, and 90% vs. 79%, respectively, showing outperformance of the predictive model. Similar results were obtained for the sex alone." What is the significance of the AUC, the OR, and specificity being listed in one sentence? The authors should clearly state the metric they are using to judge superiority of performance and justify this. Similar critique for Line 177, Line 441.
- The conclusion is unclear about the actual benefits of AI - it is not objective, as it is only as good as the training data it is developed on with all of the biases inherent in the data and the labeling of that data. Line 488 should be revised in light of that.
Author Response
Dear reviewer,
We would like to thank you for taking the time and reviewing our manuscript and providing valuable comments for further improvements. We have updated the manuscript, as suggested.
- Deep learning is not clearly described as a separate branch of machine learning that involve neural networks - these can be used for either supervised or unsupervised learning (e.g. autoencoders).
We would like to thanks the reviewer for their thoughtful comment. We have updated the manuscript as below:
A deep learning system accepts multiple data types as input, which form layers of data from which it extracts the data points of interest. Deep learning can be supervised or unsupervised. The most common models are trained using supervised learning, in which datasets are composed of input data and corresponding output data labels (6). DL comprises of two steps: pre-training and fine-tuning. In step one, the DL model attempts to learn the underlying data distribution and creates outputs in an unsupervised manner. In step two, the output generated is tuned for the specific task at hand to achieve maximum performance (5).
- Minor grammatical changes throughout that would benefit from a skilled editor: examples (not limited to these) include awkward phrasing with Line 114 "However, the small size or flat polyps could be missed by the naked eyes." - should be "could be missed by the human eye alone" or "could be missed by the naked eye". Line 126: "Thus with CADe, the risk of developing colon cancer can decrease as the ADR increases when compared with routine colonoscopy." should be more clear with "With CADe, the evidence suggests that increased ADR is achieved compared with routine colonoscopy, which may lead to reduced rates of colon cancer." ADR is a proxy measurement, and does NOT mean that the rates of colon cancer will decrease. Line 219 " However, most of the data available from small studies, so further randomized controlled studies are needed." Should be corrected with a verb, "most of the studies use data that are not from prospective clinical trials..." Line 288 is a run-on sentence. Line 295 sentence does not have a verb.
We want to thank the reviewer for their comment. We have updated the entire manuscript after giving it a thorough read. The manuscript was also scanned though a language-improving software.
- The significance of results in context are still not clearly changed. For example, Line 155 "While comparing this model to age alone, AUC, Odds ratio, and specificity were 0.81 vs. 0.72, 32 vs. 2, and 90% vs. 79%, respectively, showing outperformance of the predictive model. Similar results were obtained for the sex alone." What is the significance of the AUC, the OR, and specificity being listed in one sentence? The authors should clearly state the metric they are using to judge superiority of performance and justify this. Similar critique for Line 177, Line 441.
We have updated the manuscript as suggested by the reviewer.
- The conclusion is unclear about the actual benefits of AI - it is not objective, as it is only as good as the training data it is developed on with all of the biases inherent in the data and the labeling of that data. Line 488 should be revised in light of that.
We have updated the conclusion as suggested by the reviewer as below:
However, several study limitations need to be kept in mind. Nevertheless, most of the data is available based on small studies at tertiary care centers. Selection bias of the images used for the training set, lack of randomization of many studies, and the number of images used for the AI model's training set.
Again, we would like to thank everyone who was involved in the review of this manuscript. We hope the updated manuscript will be satisfactory to the editorial and reviewer board for the publication.
Reviewer 2 Report
Dear Authors,
I have carefully read the R1_manuscript by Dr. Hemant Goyal, et al. entitled “Scope of Artificial Intelligence in Screening and Diagnosis of Colorectal Cancer”.
Authors have clearly addressed all the previous comments.
Consistently, I have no additional relevant remark.
Minor comment
Within the polyp detection chapter, the following paragraph is related to characterization rather than to detection, please check and revise:
“Most small colorectal polyps are hyperplastic with little to no risk of turning to CRC, so an 246 accurate diagnosis of these polyps is needed to prevent unnecessary resections and complications 247 associated with it. In a single-center study from Japan, a unique CNN system based on CAD utilizing 248 AI was developed to study endoscopic images extracted from colonoscopy videos (37). A total of 1200 249 images from colonoscopies were included, and additional video images from 10 cases were applied 250 as a test. The accuracy of the 10-fold cross-validation test was found to be 0.751, meaning the decision 251 by CNN was correct in 7 of 10 cases (37).”
Author Response
Dear reviewer,
We would like to thank you for taking the time and reviewing our manuscript and providing valuable comments for further improvements. We have updated the manuscript, as suggested.
I have carefully read the R1_manuscript by Dr. Hemant Goyal, et al. entitled “Scope of Artificial Intelligence in Screening and Diagnosis of Colorectal Cancer”.
Authors have clearly addressed all the previous comments. Consistently, I have no additional relevant remark.
Many thanks for your positive comments.
Minor comment
Within the polyp detection chapter, the following paragraph is related to characterization rather than to detection, please check and revise:
“Most small colorectal polyps are hyperplastic with little to no risk of turning to CRC, so an 246 accurate diagnosis of these polyps is needed to prevent unnecessary resections and complications 247 associated with it. In a single-center study from Japan, a unique CNN system based on CAD utilizing 248 AI was developed to study endoscopic images extracted from colonoscopy videos (37). A total of 1200 249 images from colonoscopies were included, and additional video images from 10 cases were applied 250 as a test. The accuracy of the 10-fold cross-validation test was found to be 0.751, meaning the decision 251 by CNN was correct in 7 of 10 cases (37).”
Thanks for bringing our attention to this error. We have revised the manuscript as suggested.
Again, we would like to thank everyone who was involved in the review of this manuscript.
We hope the updated manuscript will be satisfactory to the editorial and reviewer board for the publication.
Round 2
Reviewer 1 Report
The authors have addressed the major points requiring revision.
This manuscript is a resubmission of an earlier submission. The following is a list of the peer review reports and author responses from that submission.
Round 1
Reviewer 1 Report
To the Authors:
This is a detailed review of artificial intelligence approaches to improve colorectal cancer screening, detection of polyps, and classifying polyps.
While this is a strong effort to organize advances in this field, there are several issues with the manuscript that limit its potential usefulness.
- The general utility of this particular review is unclear. For the clinical audience, this is a very technological jargon-heavy article without a clear message about relevant questions. From a practitioners' perspective, the important questions are: how will this help my endoscopic practice? where will the technology "enhance" my performance? From a systems perspective, the important questions are: how does an algorithm improve identification of patients who may require screening or surveillance colonoscopy? The article provide a high level of detail and studies, but no clear framework that answers these questions.
- Results overwhelm the article without necessary context. The manuscript is filled with unclear wording and unnecessary detail that obscure the central messages - in particular, the rush to present all relevant results increase confusion. An example is on line 137-138 "While comparing this model to age alone, AUC, Odds ratio, and specificity were 0.81 vs 0.72, 32 vs. 2, and 90% vs 79%, respectively...". Why is specificity important to include, and not sensitivity or positive predictive value? What does any of these differences mean in context to the question of identifying patients with potential colorectal cancer?
- The description of machine learning and deep learning is inadequate. In the introduction, the authors neglect to describe reinforcement learning, a separate branch of machine learning from supervised and unsupervised learning. Deep learning is a general term describing the use of neural networks, which can be feedforward, convolutional, or recurrent depending on the architectural configuration. "Deep neural network" is nonspecific and unhelpful.
- The inclusion of Laser-induced fluorescence spectroscopy and autofluorescence endoscopy is unclear why this is considered "artificial intelligence". It appears to be a device that collects additional data and classifies this, but does not actually employ algorithmic processing of the data per se. If this is under the "physical" AI in medicine, this should be justified.
Reviewer 2 Report
Dear Authors,
I have carefully read the review article by Dr. Hemant Goyal, et al. entitled “Scope of Artificial Intelligence in Screening and Diagnosis of Colorectal Cancer”.
This narrative review article summarizes the main evidence on different AI technologies in the field of CRC screening and diagnosis.
The authors describe the most relevant breakthroughs in this field with an analytical point of view. In some setting, the Authors also provide their personal critical and practical point of view, hereby tracing the road for the awaited new ones within the next few years.
The field of investigation is receiving tremendous attention given its emerging impact in both the endoscopic and the clinical management.
The format is entirely new. However, there is a place and need for an updated and clear overview on this topic in rapid progress.
The manuscript is clear and fluent. No table/figure has been included. The reference list appears updated and complete.
Major Comments
- Chapter 3. Polyp detection
Please note that this chapter also includes studies on polyp characterization.
Please clearly distinguish these two aspects in the text, thereby discussing evidence and open issues about polyp characterization only into the dedicated chapter.
- Chapter 4. Polyp characterization
Consistent to the previous comment, please consider starting the chapter on polyp characterization with a paragraph dedicated to WLI-driven AI systems.
Minor comments
- As a general comment
I suggest introducing a brief comment at the end of in each main chapter to highlight the Authors’ personal critical and practical point of view, hereby tracing the road for the awaited new ones within the next few years
- Please specify the term "narrative" (or descriptive) review when introducing your work in the abstract (line 38, page 1) and in the introduction (line 82, page 2).
- Endocytoscopy
The range of ultra-magnification provided on line 349, page 8 does not embrace all the EC versions, please revise (e.g., doi: 10.21037/tgh.2019.11.12) Methods
- Please consider adding one table summarizing the main AI systems (including references), the corresponding imaging technique (WLI, NBI, CLE,..) their drawback (e.g., AI system trained on fixed images only) and future implementation needed before clinical application for either polyp detection and characterization.